# Passive AI Detection of Stress and Burnout Among Frontline Workers

**DOI:** 10.3390/nursrep15110373

**Published:** 2025-10-22

**Authors:** Rajib Rana, Niall Higgins, Terry Stedman, Sonja March, Daniel F. Gucciardi, Prabal D. Barua, Rohina Joshi

**Affiliations:** 1School of Mathematics, Physics and Computing, University of Southern Queensland, Ipswich, QLD 4300, Australia; 2Mental Health and Specialist Services, The Park, Wacol, QLD 4076, Australia; 3School of Business, University of Southern Queensland, Ipswich, QLD 4300, Australia; 4Centre for Health Research, University of Southern Queensland, Ipswich, QLD 4300, Australia; 5Curtin School of Allied Health, Curtin University, Perth, WA 6102, Australia; 6School of Population Health, University of New South Wales, Sydney, NSW 2033, Australia

**Keywords:** artificial intelligence, automated stress screen, burnout, deep learning, frontline workers, mental health, occupational health, passive data

## Abstract

**Background**: Burnout is a widespread concern across frontline professions, with healthcare, education, and emergency services workers experiencing particularly high rates of stress and emotional exhaustion. Passive artificial intelligence (AI) technologies may provide novel means to monitor and predict burnout risk using data collected continuously and non-invasively. **Objective**: This review aims to synthesize recent evidence on passive AI approaches for detecting stress and burnout among frontline workers, identify key physiological and behavioral biomarkers, and highlight current limitations in implementation, validation, and generalizability. **Methods**: A narrative review of peer-reviewed literature was conducted across multiple databases and digital libraries, including PubMed, IEEE Xplore, Scopus, ACM Digital Library, and Web of Science. Eligible studies applied passive AI methods to infer stress or burnout in individuals in frontline roles. Only studies using passive data (e.g., wearables, Electronic Health Record (EHR) logs) and involving healthcare, education, emergency response, or retail workers were included. Studies focusing exclusively on self-reported or active measures were excluded. **Results**: Recent evidence indicates that biometric data (e.g., heart rate variability, skin conductance, sleep) from wearables are most frequently used and moderately predictive of stress, with reported accuracies often ranging from 75 to 95%. Workflow interaction logs (e.g., EHR usage patterns) and communication metrics (e.g., email timing and sentiment) show promise but remain underexplored. Organizational network analysis and ambient computing remain largely conceptual in nature. Few studies have examined cross-sector or long-term data, and limited work addresses the generalizability of demographic or cultural findings. Challenges persist in data standardization, privacy, ethical oversight, and integration with clinical or operational workflows. **Conclusions**: Passive AI systems offer significant promise for proactive burnout detection among frontline workers. However, current studies are limited by small sample sizes, short durations, and sector-specific focus. Future work should prioritize longitudinal, multi-sector validation, address inclusivity and bias, and establish ethical frameworks to support deployment in real-world settings.

## 1. Introduction

Burnout is a work-related condition marked by emotional exhaustion, detachment from work, and a reduced feeling of personal achievement due to long-term exposure to job stressors [1]. Although historically examined in healthcare professionals (HCPs), burnout is increasingly prevalent across all frontline sectors including emergency services, education, retail, and transportation where workers routinely encounter high workloads, emotionally demanding environments, and limited recovery time. In healthcare alone, burnout affects 35% to 54% of nurses and physicians, and between 45% and 60% of medical students and residents in the United States [2]. Similar or higher rates have been documented among teachers, police officers, and customer-facing retail staff worldwide [3,4]. A 2024 meta-analysis of 215,787 public health workers found a pooled global burnout rate of 39%. During the COVID-19 pandemic, the rate rose to 42% [5].

Burnout has far-reaching consequences for both individuals and organizations. In healthcare, it is associated with increased rates of medical error, malpractice claims, and diminished quality of patient care [6]. In other frontline settings, burnout contributes to absenteeism, decreased productivity, safety violations, and high staff turnover [7]. Individuals experiencing burnout are at increased risk of depression, cardiovascular disease, substance misuse, suicidal ideation, and overall poor quality of life. The COVID-19 pandemic exacerbated these risks, placing additional psychological strain on already overstretched frontline workforces [8].

While contributors to burnout are multifactorial spanning systemic, interpersonal, and individual-level factors, monitoring typically relies on subjective self-report measures. Commonly used instruments include the Maslach Burnout Inventory (MBI) [1], which assesses emotional exhaustion, depersonalization, and reduced personal accomplishment; the COVID-19 Burnout Scale, developed to capture pandemic-specific stressors [9]; and the Brief Resilience Scale (BRS) [10], which while not a direct measure of burnout, evaluates an individual’s capacity to recover from stress and is often used in related research. These tools are episodic and require scoring and interpretation against norms or cut-offs. Instruments like the Burnout Assessment Tool (BAT) and MBI involve complex modeling and dimensional analysis, which can lead to ambiguity if misapplied [11]. They are labor-intensive, requiring extensive validation and interpretation processes [12], and are prone to under-reporting due to stigma or fear of repercussions, as demonstrated by the development of the Burnout Stigma Inventory (BSI-8), which found that perceived stigma significantly affects disclosure of burnout symptoms [13]. There is growing consensus that organizations need more proactive, continuous methods to detect stress and burnout before symptoms escalate [14,15,16].

Recent advances in artificial intelligence (AI), wearable technologies, and digital infrastructure have enabled the development of passive monitoring systems that can detect early signs of stress and burnout [17,18]. These systems leverage a combination of physiological data (e.g., heart rate variability, skin conductance), behavioral data (e.g., physical activity, sleep patterns, mobility), and workflow behaviors. They focus on the patterns of how individuals interact with digital tools and perform tasks during their workday.

Workflow behaviors can be measured using digital traces. These may include email and message response times, calendar usage, task switching frequency, and the timing and duration of work sessions. These patterns reflect the rhythm and structure of an individual’s work and can reveal subtle shifts in cognitive load, emotional fatigue, or disengagement. For instance, reduced interpersonal communication, disrupted sleep, or changes in work rhythms may serve as non-invasive indicators of psychological strain.

By integrating these diverse data streams, AI systems can uncover complex, multi-dimensional patterns that may be predictive of burnout, patterns that are often invisible when relying on any single type of measure alone. This holistic approach enables more timely and accurate insights into employee wellbeing, potentially informing interventions before more serious outcomes emerge.

A growing body of research supports the feasibility of applying passive AI and machine learning techniques to biometric and behavioral data for the detection of acute stress [19,20,21] and mood disorders [22]. A recent scoping review of healthcare professionals (10 cohort studies, primarily using wrist-worn devices) found that heart rate and HRV were reliably associated with acute stress, and that sleep and activity metrics correlated with depression. However, no single biomarker was found to definitively predict long-term burnout [23]. The review concluded that wearable devices remain a promising yet underdeveloped tool for proactive burnout prediction.

To date, no comprehensive review has synthesized how passive AI technologies are being applied specifically for burnout detection across frontline workforces being those individuals who engage directly with clients, patients, or the public in high-demand, high-stress environments. This includes professionals in healthcare, emergency services, education, social care, and customer-facing roles. These workers often experience elevated risks of psychological strain due to the intensity, unpredictability, and emotional demands of their roles. In addition, important challenges remain unaddressed, including concerns about model accuracy, generalizability across sectors, algorithmic bias, ethical safeguards, and integration into real-world organizational workflows. Notably, while wearable devices have been a focal point in much of the research, they may not represent a sustainable long-term solution. Frontline workers often discontinue device use due to discomfort, privacy concerns, or lack of perceived benefit [24].

To address these gaps, we conducted a narrative review of passive AI approaches for detecting stress and burnout in frontline occupations. Recognizing the limitations of wearable-only monitoring, which often captures a narrow range of physiological signals and may miss contextual or behavioral nuances the focus has been broadened to include four complementary methods: (a) communication pattern analysis, (b) biometric and ambient sensing, (c) workflow analytics, and (d) sensor fusion. This expanded scope is important because burnout is a complex, multi-dimensional phenomenon that cannot be fully understood through physiological data alone. For example, changes in digital communication habits, task-switching frequency, or calendar usage may signal cognitive overload or emotional fatigue that wearables cannot detect. By integrating diverse data streams, including how individuals interact with work systems and environments, passive AI systems can reveal richer, more predictive patterns of strain. This review synthesizes current applications of these technologies across frontline workforces, defined as those in direct, high-stress engagement with the general public. Our objective is to identify strategies that enable passive, low-burden detection of distress in frontline contexts. We also explore challenges related to privacy, equity, and workplace integration. Specifically, we aim to identify the physiological and behavioral signals most strongly associated with stress and burnout, highlight recent methodological advances (2020–2025), and outline the key barriers and research needs for responsible and sustainable deployment of passive AI tools to support frontline worker well-being.

Importantly, while healthcare has been a major focus of burnout research, this review also includes studies from other frontline sectors such as education, emergency services, and retail. These occupations share similar stressors—emotional labor, unpredictable workloads, and frequent public interaction—that make them suitable for passive AI monitoring. To reflect this broader scope, we clarified our eligibility criteria to include general working populations where findings are applicable to frontline roles. This cross-sectoral inclusion enhances the relevance and generalizability of our synthesis and supports the development of passive AI systems that can be adapted across diverse occupational settings.

Several literature reviews have previously examined stress and burnout detection in frontline populations. We summarize these in Table 1. Of the seven reviews identified, only one explicitly focused on frontline workers, and none examined the use of passive data sources. This highlights a clear gap addressed by the current narrative review.

## 2. Background

### 2.1. Physiology of Stress

Stressors, whether physical or psychological in nature, activate integrated neuroendocrine and autonomic responses. The sympathetic–adreno–medullary (SAM) branch of the autonomic nervous system provokes a rapid “fight-or-flight” response: the adrenal release of epinephrine and norepinephrine increases heart rate, blood pressure, and glucose mobilization, while suppressing non-essential functions such as digestion [31,32].

The hypothalamic–pituitary–adrenal (HPA) axis provides a slower hormonal response: stress triggers hypothalamic CRH release, stimulating pituitary adrenocorticotropic hormone (ACTH) and ultimately cortisol secretion [33,34]. Cortisol follows a circadian rhythm (high in the morning, low at night), but stress-induced HPA activation causes cortisol elevations that increase cardiorespiratory output [34].

Sleep disruption, fatigue, and cognitive effects are hallmark behavioral outcomes of stress. Elevated cortisol and sympathetic activity reduce sleep quality: chronic insomnia is linked to 24 h hypersecretion of ACTH and cortisol, indicative of HPA overdrive [35]. Burnout, a consequence of chronic stress, is strongly associated with sleep loss and daytime fatigue, with HPA/SAM dysregulation underlying both [36].

### 2.2. Stressors in Modern Work

Modern occupational demands can intensify stress physiology. Work overload stimulates both the HPA and SAM axes. For instance, healthcare workers exposed to pandemic-era work surges showed elevated salivary cortisol and self-reported stress symptoms in real-time diaries [37].

Emotional labor in customer service or healthcare, where individuals modify or fake their emotional expressions without changing their actual internal feelings, depletes emotional resources, thereby accelerating exhaustion [38]. Shift work and irregular schedules disrupt cortisol’s circadian rhythm, leading to impaired sleep, fatigue, and reduced cognitive alertness, particularly during night shifts [39,40].

Lastly, digital hyper-connectivity sustains chronic stress through techno-overload, where individuals are pressured to work faster and manage excessive information, and techno-invasion, which blurs boundaries between work and personal life by fostering a constant state of connectivity. Studies show that increased technostress during the COVID-19 pandemic correlated with reduced sleep duration and greater mental fatigue [41].

## 3. Method

### 3.1. Data Sources and Search Strategy

We conducted a narrative review of peer-reviewed literature on passive AI approaches for detecting stress and burnout among frontline workers. Searches were carried out on 10 June 2025, with no date restrictions, across five major bibliographic platforms: PubMed, IEEE Xplore, Scopus, ACM Digital Library, and Web of Science. We combined controlled vocabulary (e.g., MeSH terms in PubMed, Thesaurus terms in IEEE Xplore) with free-text keywords related to “burnout,” “stress,” “wearable,” “sensor,” “electronic health records,” “communication logs,” “organizational network analysis,” and “artificial intelligence.”

This review employed a narrative synthesis rather than a systematic review due to the substantial heterogeneity in study designs, data modalities, and outcome measures across the literature. Included studies varied widely in terms of occupational settings (e.g., healthcare, education, emergency services, retail), passive data sources (e.g., wearable sensors, electronic health record logs, communication metadata), and AI methodologies (e.g., support vector machines, random forests, neural networks, federated learning). Such diversity precluded meaningful statistical aggregation or meta-analysis.

This narrative approach enabled broader exploration of emerging trends, conceptual frameworks, and implementation challenges in passive AI–based stress and burnout detection. It also allowed inclusion of early-stage, pilot, and conceptual studies that may not meet the strict methodological criteria of systematic reviews but offer valuable insights into feasibility, innovation, and sector-specific applications.

To ensure methodological transparency and rigor, the Mixed Methods Appraisal Tool (MMAT) was applied to all included studies. This structured evaluation supports the interpretive synthesis by contextualizing findings within the quality and limitations of the evidence base. Future research may benefit from a formal systematic review once the field matures and a more standardized body of literature becomes available.

Studies were selected based on their relevance to real-world challenges such as privacy protection, demographic generalizability, and ethical considerations. Collectively, these criteria contributed to a comprehensive and analytically balanced review, offering critical insights into both the capabilities and limitations of passive AI systems for monitoring mental health in occupational settings.

### 3.2. Review Strategy

All records retrieved were imported into EndNote X21 for de-duplication. Two reviewers (NH and RR) independently screened titles and abstracts against the following inclusion criteria:Population: Frontline workers (healthcare professionals, educators, emergency responders, retail staff), or general working populations where findings are applicable to frontline roles.Data Type: Passive data sources (wearable sensor outputs, EHR audit logs, digital communication metadata, ambient environmental sensors).Outcome: Measures or inferences of stress or burnout.Study Design: Any design where AI or machine-learning methods have or could be applied to the above data for stress/burnout detection.

Studies solely relying on self-reported or active measures (e.g., survey only designs) were excluded. Discrepancies at the title and abstract stage were resolved by discussion; studies meeting initial criteria underwent full text review by the same paired reviewers. An investigator (SM) adjudicated any remaining disagreements.

### 3.3. Extraction Strategy

A standardized data extraction form was piloted on five randomly selected papers and refined for clarity. One investigator (NH) extracted the following details from each included study:Study setting and population characteristics (sample size, profession, demographics)Passive data modality (e.g., HRV, skin conductance, EHR clickstreams, email metadata, ambient sensors)AI/machine-learning techniques employed (e.g., Support Vector Machine (SVM), random forest, neural networks, federated learning (FL))Outcome measures (burnout scales, stress questionnaires) and reference standardsDuration of data collection and timing of assessmentsReported performance metrics (accuracy, F1 score, AUC)

One investigator (RR) reviewed extracted data for accuracy; any discrepancies were resolved by consensus. As shown in Figure 1, the PRISMA flow diagram summarizes the identification, screening, and inclusion process for studies in this review.

Table 2 summarizes a select illustrative sample of recent studies (2020–2025) on passive AI-based stress and burnout detection. To clarify the scope of studies reviewed, we note that Table 2 presents a select illustrative sample chosen for methodological rigor and relevance to the review’s objectives. The full set of included studies is described throughout the main text. A total of 10 studies met the inclusion criteria and were assessed using the Mixed Methods Appraisal Tool (MMAT) to support structured evaluation of methodological quality. To provide a consistent lens for evaluating methodological rigor, the Mixed Methods Appraisal Tool (MMAT) was applied to each study, where applicable. The inclusion of quality scores is intended to help contextualize findings, especially given the variability in study design and sample sizes. This approach blends elements of narrative review with structured appraisal, offering a balanced overview of the field. Future work may benefit from a formal systematic review to more comprehensively assess the evidence base.

## 4. Literature Reviews on AI-Based Stress and Burnout Detection

### 4.1. Performance Comparisons of Various Methods

The studies included in this review were chosen based on their methodological strength, focused on relevant job sectors, and their contribution to addressing existing knowledge gaps. This review focused on studies that utilized passive data such as from body sensors, work activity logs, and digital communications to detect stress and burnout, thereby avoiding the need for traditional surveys. By avoiding traditional survey-based approaches, this method supports continuous, unobtrusive monitoring that is less susceptible to bias and data loss.

Recent studies have demonstrated moderate to high classification accuracy in detecting stress and burnout using AI-based methods. However, results vary significantly depending on context, data modality, and methodology. Overall, machine learning models typically achieve 75-90% accuracy or F1 scores in small-scale tests [27,43], but performance may decline in more diverse, real-world deployments. Importantly, no single “best” sensor or algorithm has emerged and has led to adoption of ensemble approaches applied to multimodal data that often achieve better outcomes [27]. Many published trials are short-term or retrospective, highlighting the need for long-term prospective validation.

The included studies span diverse frontline occupations, including healthcare, education, emergency services, and retail, thereby capturing the unique stressors and data environments of high-demand public-facing roles. Furthermore, the selection aimed to represent a broad spectrum of AI methodologies from support vector machines and random forests to federated learning and generative models, illustrating the evolving technological landscape.

### 4.2. Communication Pattern Analysis

This approach uses digital communication logs (email, messaging, calls) to infer stress. Selected studies have analyzed various communication features like message frequency, length, response time, and sentiment. Estévez-Mujica’s (2018) study of 57 R&D employees (52,000 emails) found that email-based features could explain up to 34% of variance in burnout scores; a machine learning classifier achieved an F1 score of 0.84 (100% recall, 73% precision) in distinguishing high-risk employees [43]. Key predictors included late night and weekend emailing, lack of social reciprocity, and negative sentiment. These findings suggest that increased communications (especially times outside regular working hours) are markers of exhaustion.

A systematic review of healthcare-related stress detection methods conducted by Pinge et al. found that patterns related to increased secure messaging and after-hours replies are associated with physician stress. The key advantage of these approaches is that passively collected data is used which minimizes the need for additional hardware. These methods have the advantage of using already-collected data with minimal additional hardware. Performance metrics (accuracy, recall/F1) vary by dataset, but high classification scores (often 75–85%) are reported in small studies [27,43]. Critiques include privacy concerns (monitoring communications) and potential false positives (e.g., a very busy but not burned-out person). Notably, most of the identified studies in this review were conducted office or hospital settings; evidence from cross-industry sectors like retail or hospitality messaging platforms is still emerging.

### 4.3. Organizational Network Analysis

Organizational Network Analysis (ONA) examines patterns of collaboration and support within teams. AI can map who talks to whom (via email, chat, meeting logs) and detect changes (e.g., isolation of individuals) [51]. For example, a lapse in communication during care coordination can lead to inadequate patient follow-up, delayed care and provider burnout. Although no large empirical studies have yet tested ONA for burnout detection specifically, many vendors promote ONA dashboards for well being. Both “bridge” individuals, those who connect otherwise isolated teams and “silos” which are groups with limited external communication, can signal potential risks for burnout. Research on ONA in well being is nascent; one needs to ensure that network metrics align with actual stress measures. A potential critique is that ONA assumes formal communication reflects support—it may miss informal stress or external factors [27]. In practice, ONA tools must also handle privacy (mapping connections can be sensitive) and often have not been rigorously validated for predictive power.

### 4.4. Biometric Sensing

The approach studied the most has been passive data collection from wearable sensors. Common data include heart rate (HR), heart rate variability (HRV) [44], skin conductance (EDA), skin temperature, movement, and sleep patterns. Wearables can detect physiological stress responses (sympathetic arousal) and correlate them with burnout [27]. Li, Zhu, Sui, Zhang, Chi, and Lv studied 17 nurses using wearable ECG patches to measure HRV indices [44]. The HRV features significantly correlated with nurses’ self-rated stress scores [44]. They suggested that work shifts (e.g., night vs. day shifts) influence stress levels in measurable ways. Another review found many studies using Random Forest or SVM on multi-sensor data achieved accuracies of 80–95% for detecting stress states with one decision-tree model reaching 95% accuracy on a benchmark stress dataset [27]. These are typical of pilot studies conducted to reinforce feasibility.

Biometric sensing excels at capturing real-time physiological changes and is truly passive once worn. But there are limitations: sensor noise, motion artifacts, and wearer compliance (devices must be worn consistently). Studies often use small samples or controlled stressors; generalizing to diverse work environments is ongoing. Wearables may also perform differently on different users (e.g., optical HR sensors can be less accurate on darker skin). Integration with contextual data (shift schedules, patient load) is currently limited, though combining wearables with administrative data (e.g., patient acuity) could improve predictions [27]. The model described by Pinge, Gad, Jaisighani, Ghosh, and Sen outlines a wearable-based stress detection framework that integrates three key data sources: physiological signals collected passively via wearable devices (e.g., heart rate, heart rate variability, sleep), contextual and environmental factors (e.g., workload, shift patterns), and self-reported mood or stress levels gathered through ecological momentary assessments [27]. These multimodal inputs are processed using machine learning techniques to detect stress episodes and monitor trends, enabling personalized and potentially real-time interventions for stress and burnout management.

### 4.5. Workflow Interaction Analytics

This method uses digital workflow logs to infer workload and stress. These may include electronic health record (EHR) audit trails, electronic instrument usage, or computer activity logs. The idea is that changes in how workers interact with systems (e.g., long login sessions, frequent interruptions, delayed documentation) signal overload. Tiase, Sward, and Facelli illustrate this approach in their nursing informatics model (“RNteract”) [45]. It structures EHR audit-log data into nurse tasks, patient panels, and nurse types, aiming to predict outcomes like burnout [45]. Their model treats charting, navigation, and searching patterns in the EHR as candidate burnout markers. EHR audit logs are organized by tasks (charting, order entry), nurse types (experience, role) and patient panels. AI algorithms can then mine these patterns to predict burnout [45]. This schema is extensible to include other data (e.g., survey responses, staffing) as inputs.

Empirically, workflow analytics is emerging. Studies have used EHR logs to measure workload or detect inefficiencies (e.g., average documentation time per patient). Shan et al. (2023) used nursing EHR interactions to approximate “time on task” and found it related to self-reported workload [52]. So far, predictive models directly linking workflow features to burnout scores are rare. The chief critique is that workflow data only capture part of the work (for instance, patient care or downtime are invisible). Integration of multiple data (EHR logs, wearable, staffing ratios) is an active research area. Log data also come with privacy concerns (tracking employees’ every click) and technical challenges (standardizing heterogeneous log formats).

### 4.6. Emerging Technologies

#### 4.6.1. Generative AI Support Systems

Large-language models (LLMs) and generative AI (e.g., ChatGPT) are being deployed to reduce administrative burden—an indirect way to mitigate burnout. For example, Garcia et al. (2024) found that using ChatGPT-4 to generate draft replies to patient portal messages was associated with statistically significant reductions in clinician cognitive burden and work exhaustion [46]. In a 5-week pilot involving 162 clinicians, the intervention improved measures of clinician well-being, although no significant time savings were observed. Likewise, generative AI is used to summarize EHR notes, code documentation, or triage tasks. In principle, these tools act proactively by shrinking stressors (e.g., paperwork), while passively monitoring via language (e.g., NLP analysis of clinician notes).

Generative models also open new passive-use cases: an AI assistant could monitor a clinician’s journal entries or messages for negative language patterns, or synthesize risk insights from sensor data. However, they raise concerns: hallucinations (factual errors), data privacy (models trained on sensitive text), and transparency issues. The Stanford group also recently advocated for an “AI code of conduct” in healthcare to ensure safety, inclusivity and ethics [46]. In summary, generative AI holds promise as both a support (reducing burnout causes) and an analytical tool, but demands careful human-centered integration (see 6. Design and Ethical considerations below).

#### 4.6.2. Sensor Fusion and Ambient Computing

Combining multiple sensor modalities (“sensor fusion”) can improve stress detection. For instance, pairing a wristband (physiology) with environmental sensors (noise levels, light, temperature) or smartphone usage patterns can give a fuller picture. Haghi, Danyali, Thurow, Warnecke, Wang, and Deserno developed a wrist-worn prototype that integrates physiological (heart rate, skin temperature), environmental (air quality, temperature, sound), and behavioral (motion tracking) sensors to monitor worker strain. While not explicitly framed in terms of stress or burnout, the system’s multidimensional data fusion offers a promising approach for detecting early signs of occupational stress and environmental contributors to burnout [47]. Liu et al. (2021) demonstrated that subtle physiological cues that included blink rate captured via camera and vocal tone analyzed through microphone can serve as reliable indicators of psychological stress, particularly in mobile and real-time contexts [53]. Sensor fusion offers strong theoretical potential for improving stress detection, but its implementation in real world workplace settings remains limited. Few studies have rigorously tested multi-sensor models outside controlled environments. Additionally, privacy concerns are amplified by the use of cameras and microphones which may be perceived as intrusive. However, this type of ambient computing does offer the potential for smart office environments to unobtrusively detect workforce well-being. For example, a conceptual sensor suite in a break room might infer collective tension by analyzing conversational dynamics or physical posture. While largely speculative at present, Pinge et al. (2024) have demonstrated the feasibility of both wearable and ambient systems for stress detection in industrial contexts [27]. The broader promise lies in ubiquitous sensing by combining ambient and personal data sources without requiring active user input. However, this approach presents significant challenges, including high data volumes, persistent privacy concerns due to continuous environmental surveillance, and the need for local data processing via edge computing.

#### 4.6.3. Federated and Privacy-Preserving Learning

Privacy of personal health data is critical in potential burnout detection. Federated learning (FL) offers a promising solution by enabling models to be trained locally on-device, with only model updates rather than raw data shared with a central server. This allows multiple institutions to collaboratively train stress detection models without compromising employee privacy. For example, Fauzi, Yang, and Blobel used FL in a smartwatch-based stress detection study on the WESAD dataset. While centralized personal training achieved near-perfect accuracy (99.98%), the federated model performed slightly worse [49]. Nonetheless, FL remains a compelling approach for preserving confidentiality in regulated environments.

However, FL faces technical challenges, particularly in managing non-independent and identically distributed (non-iid) data, where stress patterns vary across work settings. These variations can often be traced to differences in shift schedules, task complexity, staffing ratios, and access to support resources. Policy and operational documents including job descriptions, scheduling protocols, wellness program guidelines, and staffing reports can be valuable sources for inferring these contextual factors. Incorporating such insights may help tailor federated models to better reflect the lived realities of diverse workplaces. Additionally, FL requires frequent communication between devices to share model updates, which can slow down the system and use up network resources. This makes it harder to scale and respond quickly in real-world settings like hospitals or large organizations.

## 5. Implementation Challenges

The large-scale deployment of passive AI systems for burnout detection entails several complex and interrelated challenges that must be carefully addressed to ensure effectiveness, reliability, and ethical integrity.

### 5.1. Data Privacy & Regulation

Health and employment data are highly regulated. In healthcare settings, HIPAA (US) or General Data Protection Regulation (GDPR) European Union (EU) rules apply when AI analyses clinician data. Wearable physiological data may be deemed medical information. Organizations must secure informed consent, anonymize data, and possibly obtain Food and Drug Administration (FDA) or equivalent clearance for devices that diagnose burnout [23,46]. For instance, digital health apps used for stress/mental health may require “Software as a Medical Device” (SaMD) approval. Cross-border deployments must handle varying laws.

#### Ethical Concerns

Employees may fear surveillance, data sharing, and a lack of privacy. Transparent communication and strict limits on use (e.g., anonymized, aggregate alerts rather than individual flags) are essential. Ethical AI frameworks (fairness, accountability, transparency) should guide design. For example, a recent Journal of the American Medical Association (JAMA) viewpoint calls for AI codes of conduct ensuring safety, inclusivity, and sustainability in healthcare AI [46]. Bias is a risk: an algorithm trained on young urban nurses may not generalize to more experienced or rural workers. Ongoing auditing for bias and adaptation to context is needed.

### 5.2. Technological Limitations

Despite the promise of AI and wearable technologies in stress detection, several limitations persist. Sensors and models can fail, and wearables may drop data due to battery depletion or connectivity issues [23]. Communication logs are often incomplete, especially when teams use unsanctioned messaging platforms. AI models frequently lack robustness; for example, Fauzi, Yang, and Blobel found that the accuracy of a machine learning-based stress detection model significantly declined when tested on new data, underscoring the challenges of generalizability in federated learning (FL) contexts [49]. Integration also remains difficult as many organizations lack the IT infrastructure to channel wearable or electronic health record (EHR) data into analytics pipelines. Furthermore, interoperability standards for stress-related data are currently non-existent.

Few studies have systematically examined bias in stress detection systems. For instance, Van Zyl-Cillié, Bührmann, Blignaut, Demirtas, and Coetzee conducted research among nurses in South Africa in a resource-constrained setting, highlighting the need for cross-cultural validation [50]. Wearable devices themselves may exhibit performance disparities. For example, optical heart rate sensors have been shown to perform less reliably on darker skin tones [27]. To ensure fairness and generalizability, future research should intentionally evaluate algorithms across diverse groups, including gender, race, age, and professional roles.

### 5.3. User Trust and Workflow

Research on AI-powered Clinical Decision Support Systems (AI-CDSS) consistently shows that poor alignment with clinical workflows and lack of user trust are key reasons many tools fail to gain adoption [54]. These practical problems including misalignment with work processes, unclear decision-making, and insufficient training can lead users to ignore alerts or disable systems altogether. Successful implementation requires interpretable feedback, transparent onboarding, and robust support for training and change management.

In summary, effective implementation demands a socio-technical approach. Technical excellence alone is insufficient; leadership, policy, and culture matter. Behavioral science offers complementary insights, grounding the design and deployment of passive AI tools in evidence-based principles of human motivation, decision-making, and organizational change. For example, user interface design should follow human-centered design principles involving frontline staff early in prototyping and maintaining continuous feedback loops [54,55].

When identifying burnout, it is important to recognize that stress stems not only from individual traits but also from the structure of the work environment and the demands it imposes on employees.

Several key factors contribute to overall stress levels in the workplace. For example, patient load refers to how many patients each healthcare worker manages, and a higher load can lead to feelings of being overwhelmed. Case complexity deals with the intricacies and severity of the medical cases, as more complex situations require greater mental and emotional resources. Additionally, shift schedules can significantly impact work–life balance; irregular or excessively long shifts can disrupt personal time and increase fatigue. Staffing levels are also crucial; when staffing is inadequate, remaining staff may face greater responsibilities, intensifying their stress. Finally, access to support resources, such as mental health services and peer support networks, plays a vital role in managing stress, limited access can heighten feelings of isolation and burnout.

Factors can differ significantly across various healthcare environments, such as busy urban hospitals, small community clinics, and specialized outpatient facilities that focus on specific health services. These differences are usually outlined in job descriptions that specify the roles and responsibilities of healthcare staff, scheduling protocols that manage patient appointments and staff availability, and wellness program guidelines aimed at supporting the mental and physical health of employees. Incorporating this contextual information, especially in federated learning frameworks that enable collaborative data analysis while protecting privacy, can enhance the relevance and fairness of predictive models designed to address healthcare professional burnout. This strategy recognizes the unique challenges faced by healthcare workers in different settings and promotes the creation of tailored support systems that effectively meet their needs. By emphasizing this understanding, we can contribute to a healthier workplace culture that prioritizes the well-being of every healthcare professional. Self-determination theory helps guide the design of technology so it supports people’s basic psychological needs like feeling in control and capable as well as being connected to others [56], while organizational change literature highlights the importance of participatory approaches and leadership alignment in successful AI adoption [57].

## 6. Design and Ethical Considerations

Any AI burnout tool must be designed not only with functionality in mind but also with the well-being of workers at its core. Key principles to guide its development include:

### 6.1. Human-Centered Co-Design

Involving frontline staff, managers, and union representatives in the design of digital tools ensures alignment with real-world needs and fosters trust. Participatory design methods like co-design workshops and pilot testing are used in an effort to enable development of user-aligned tools. A recent case study in the Australian healthcare system highlighted how working together with clinicians and patient representatives to design AI-based workflows improved the integration of patient-reported experience measures (PREMs) into regular hospital quality improvement processes [55]. In the same way, collaborating with end-users to create burnout detection interfaces, like dashboards, could enhance usability, ensure contextual relevance, and promote adoption by including their experiences in the design process.

### 6.2. Transparency and Explainability

AI systems should provide outputs that are easy to understand and well-articulated. When an algorithm tags a worker as high-risk, it is important to give a clear and specific explanation for that classification. For instance, the system could point out, “We’ve noticed an up tick in late-night emails and a steady drop in physical activity over the past two weeks.” This kind of transparency encourages accountability and helps ensure effective human oversight. Plus, having override options is really important. It allows managers or HR professionals to review decisions made by the algorithm, helping to minimize the chances of unfair or incorrect outcomes. Ultimately, it is all about using AI responsibly and making sure it enhances our workplaces rather than complicating them.

### 6.3. Equity and Bias Mitigation

To promote fairness, AI systems should adhere to established ethical frameworks. for example, the FAIR (Findable, Accessible, Interoperable, Reusable) principles proactively address bias during data collection and model development. Doing so includes sourcing diverse datasets across sectors and geographies, and validating model performance across demographic subgroups [54,55]. Equally important are individual-level safeguards: systems should offer opt-out options and robust privacy controls to respect autonomy and protect sensitive information.

### 6.4. Continuous Evaluation

After deployment, it is crucial to monitor outcomes to assess real-world effectiveness. Key questions include: Does early detection of burnout lead to meaningful interventions that demonstrably reduce burnout rates? The success of such interventions should not be assumed but evaluated systematically.

Refinement of these approaches should be guided by principles of implementation science, which emphasize iterative improvement based on empirical evidence. This includes the use of pre- and post-intervention metrics, satisfaction surveys, and safety indicators to assess both process fidelity and outcome effectiveness. For example, Damschroder et al. (2009) [58] propose a consolidated framework for implementation research that can help identify contextual factors influencing success or failure. Implementation strategies specific to health service context and workers may also be required to enhance uptake of tools. Similarly, Proctor et al. (2011) [59] outline key outcomes for implementation research, including acceptability, feasibility, and sustainability, which are essential for evaluating burnout interventions in practice.

Many existing studies are limited to short-term evaluations. To establish long-term impact, extended longitudinal research that track cohorts over multiple years is needed. The BROWNIE burnout prediction study by Wilton et al. [24] exemplifies this approach. It recruited 360 registered nurses across three cohorts over a one year period, using consumer-grade smartwatches to collect continuous physiological data, alongside quarterly psychological surveys and administrative workplace metrics.

This decentralized, digital health protocol was designed to minimize participant burden and ensure inclusivity across shift types. By integrating physiological, psychological, and organizational data, the study aims to develop predictive models of burnout and estimate associated costs. Its use of probabilistic graphical models and multitask learning sets a new benchmark for predictive research in healthcare, offering a scalable framework for early intervention and institutional response.

This review makes a distinctive contribution by synthesizing evidence on passive AI-based detection of stress and burnout across a variety of frontline sectors, including healthcare, education, retail, and emergency services. Unlike previous reviews that concentrate on wearable technologies or focus on individual sectors, our analysis combines findings from multiple passive data sources, including biometric signals, workflow metrics, and digital communications. Together, these sources demonstrate strong potential for identifying early signs of psychological strain.

Furthermore, this review identifies critical gaps in the literature, including the lack of cross-sector generalizability, limited longitudinal validation, and insufficient attention to ethical governance and integration into organizational workflows. By addressing these gaps, our work offers actionable insights for researchers and practitioners aiming to develop scalable, equitable, and ethically sound AI-driven monitoring systems for workforce well-being.

## 7. Gaps and Future Directions

Although the body of literature is expanding, several important gaps persist:

### 7.1. Cross-Sector and Cross-Cultural Research

Current studies predominantly examine hospital staff in high-income countries, leaving retail, food service, transportation, and the military under-represented. Moreover, cultural influences, including attitudes toward work and technology, likely shape both burnout experiences and the data signals used to detect them. To address these limitations, future research should prioritize the collection and ethical sharing of datasets from a broader range of industries and geographic regions.

### 7.2. Longitudinal Designs

There remains a significant gap in large-scale, longitudinal research on passive stress monitoring, particularly regarding its potential to prevent burnout. A central question is whether continuous stress detection can serve as an effective early warning system to mitigate burnout risk. Addressing this requires multi-year investigations similar to that employed by the BROWNIE study [24] alongside targeted intervention trials, including “just-in-time” alerts. These studies could include comparison groups and use clear, measurable results like days off work, staff attrition, and healthcare costs to properly assess their impact.

Traditional longitudinal research designs are effective for examining long-term developmental and behavioral changes. However, they may lack the temporal sensitivity required to detect short-term variations, which are essential for understanding the dynamics of stress. To address this limitation, measurement burst designs have been proposed as a complementary approach. These designs incorporate intensive ecological momentary assessments that may take multiple daily measurements over say a two-week period. A broader longitudinal framework could possibly be spaced at intervals of several months. This type of methodology could enable the simultaneous capture of acute stress responses and chronic stress trajectories. An applied example of this approach is presented in Reubenson et al. (2025) [60].

### 7.3. Intercultural Data Challenges

Emotional expression and communication styles differ significantly across cultures. AI models trained primarily on Western populations may misinterpret behavioral patterns from Asian, Middle Eastern, or other cultural contexts. Advancing multilingual and culturally sensitive natural language processing (NLP) is a key area for future research. Occupational role can significantly influence both stress perception and the type of data available for monitoring. For example, retail workers might not use email much. To ensure robustness, models must be evaluated across diverse workplace environments.

### 7.4. Outcome Validation

Many studies rely on self-reported burnout scales as their primary outcome. However, fewer have established links between passive data signals and objective outcomes. Future research should aim to validate predictive models against concrete endpoints such as accident rates, absenteeism, or health outcomes to better demonstrate practical utility.

### 7.5. Policy and Ethical Governance

Researchers should collaborate with policymakers to establish regulatory frameworks tailored to worker-monitoring AI. These frameworks could draw inspiration from existing standards like the GDPR and the forthcoming EU AI Act. Clear guidance is needed on issues of informed consent, data ownership, and acceptable interventions to ensure ethical deployment and to safeguard worker rights.

## 8. Discussion

Wearable sensing, workflow analytics, and communication pattern analysis are some examples of technology used to detect and predict burnout, stress, and related forms of distress among frontline workers. Across the studies reviewed, wearable devices most frequently collected data on heart rate (HR), heart rate variability (HRV), electrodermal activity (EDA), temperature, sleep, and motion. These deployments ranged from brief, single-shift monitoring to multi-week studies involving small cohorts of healthcare professionals and volunteers.

As this is a narrative review, no meta-analysis was conducted. A statistical synthesis was not feasible due to substantial heterogeneity in study designs, populations, data sources, and outcome measures across the included literature. Instead, we applied the Mixed Methods Appraisal Tool (MMAT) to provide a structured and transparent assessment of methodological quality. This approach allowed us to contextualize findings while acknowledging the variability in evidence strength and study rigor.

Across the included studies, several consistent sources of bias were identified. Wearable-based studies frequently exhibited selection bias due to small, self-selected samples, often drawn from single institutions. Measurement bias was also common, with sensor noise, motion artifacts, and inconsistent device adherence affecting data quality. Notably, optical sensors may perform less reliably on darker skin tones, raising concerns about racial bias. Additionally, wearable studies often lacked contextual data such as shift schedules or workload, limiting interpretability of physiological signals.

Workflow-based studies presented different bias risks. Selection bias arose from reliance on data from specific electronic health record (EHR) systems, which may not generalize across platforms. Measurement bias was introduced through incomplete or misinterpreted log data, as not all work activities are digitally captured. Privacy concerns and reactivity bias were also noted where users altered their behavior due to awareness of monitoring.

Few studies reported power calculations or conducted subgroup analyses to assess algorithmic fairness. Only one study explicitly addressed demographic generalizability. These limitations underscore the need for future research to incorporate larger, more diverse samples, longitudinal designs, and structured bias auditing.

Recent studies have explored HRV as a physiological marker of stress, with at least three investigations reporting on its use [44,61,62]. However, only one study demonstrated a strong correlation between HRV and real-world stress outcomes. This finding aligns with previous research conducted outside of healthcare settings, which observed changes in heart rate variability (HRV) under controlled, short-term stress conditions [63]. However, lab-based studies often provide only brief snapshots of physiological responses and may not accurately reflect the varied stress patterns experienced in real-world workplaces.

To improve our understanding of long-term stress and burnout, future research should focus on longer monitoring periods. This could include intensive assessments, like ecological momentary assessments or daily evaluations, along with long-term studies lasting several months or even years. It would also be beneficial to pair physiological data with validated psychological tools, such as the Maslach Burnout Inventory (MBI) [1], the Patient Health Questionnaire-9 (PHQ-9), and the Generalized Anxiety Disorder-7 (GAD-7). Additionally, it is important to conduct sector-specific analyses, as stress profiles can differ significantly between various professions, such as hospital nurses and retail workers, or between urban and rural settings.

As frontline work becomes increasingly digital and decentralized, low-burden, app-based passive monitoring tools offer promising avenues. These systems should ideally provide real-time feedback to enhance user engagement and ethical use. Key ethical concerns including informed consent, perceptions of surveillance, and comfort with physiological tracking must be addressed to ensure equitable adoption. Discrepancies between wearable-derived metrics and subjective fatigue, particularly regarding sleep and cognitive load, also warrant further investigation.

In Australia, psychological claims for workers compensation typically require a diagnosable mental disorder, often framed as a workplace ‘injury’ [64]. The costs associated with these claims for organizations is increasing and subsequently there is increasing pressure to tighten eligibility criteria [65,66]. Burnout, while not always formally diagnosed, is likely a significant contributor to these trends. The passive AI approaches discussed in this review may help organizations to identify symptoms that may lead to burnout earlier and support worker wellbeing proactively.

Passive AI signals offer significant potential for integration into existing clinical and organizational workflows. For example, biometric data from wearables could be linked to electronic health records (EHRs) to provide clinicians with real-time indicators of psychological strain. Workflow analytics, such as EHR usage patterns or documentation delays, could be embedded into clinical dashboards to flag operational stressors. Communication metrics, including email timing and sentiment, may be used by managers to monitor team well-being without requiring self-reporting.

To operationalize passive monitoring for preventive intervention, organizations could implement tiered alert systems that trigger support actions based on risk thresholds. For instance, a sustained drop in sleep quality combined with increased after-hours digital activity might prompt a confidential check-in from a supervisor or automated referral to employee assistance programs. These systems should be designed to deliver aggregate insights at the team or unit level to avoid stigmatization and preserve privacy.

Integration into workflows also requires alignment with existing clinical governance structures. Passive AI outputs should be interpretable, actionable, and embedded within routine decision-making processes. For example, burnout risk scores could be incorporated into shift planning tools, allowing managers to adjust workloads or offer recovery time. In clinical settings, stress indicators could inform supervision intensity or peer support allocation.

Ultimately, the success of passive AI systems depends on their ability to complement human judgment, not replace it. Co-design with frontline staff, clear communication about data use, and robust ethical safeguards are essential to ensure trust and uptake. When thoughtfully implemented, passive monitoring can shift burnout detection from reactive to proactive, enabling earlier interventions and fostering healthier work environments.

This review has several limitations. Most included studies focused on healthcare professionals, with limited representation from other frontline sectors such as retail, law enforcement, education, and call centers. Participant demographics, race, gender, and ethnicity were often under-reported, and few studies included under-represented groups. A recurring limitation across the included studies was small sample size, which constrains statistical power and limits the generalizability of findings. Many wearable-based studies, for example, involved fewer than 50 participants and were often conducted within a single institution, increasing the risk of selection bias. These small cohorts also reduce the ability to detect subgroup differences or validate algorithmic fairness. In contrast, the study by Van Zyl-Cillié, Bührmann, Blignaut, Demirtas, and Coetzee included a substantially larger sample of 1165 nurses, representing a relative strength within the evidence base [50]. Although this study relied on survey data rather than passive sensing, its scale demonstrates the feasibility of recruiting large frontline cohorts for AI-driven burnout research and underscores the need for similar efforts in passive monitoring studies. Given known disparities in digital health access and occupational stress, future research must establish standardized protocols, address ethical governance, systematically evaluate bias and fairness and validate predictive models across diverse frontline roles.

## 9. Conclusions

Passive AI tools like wearables, communication logs, and workflow data can help spot early signs of stress and burnout in frontline workers without needing them to fill out surveys. These systems can track changes in sleep, heart rate, work habits, and communication patterns to give early warnings before things get worse.

Most current studies focus on healthcare and are short-term, so more research is needed across different jobs, cultures, and over longer periods. Privacy and fairness are also important as workers need to trust that their data is safe and used to help, not punish.

When developed carefully and used thoughtfully, passive AI can significantly enhance the workplace by providing timely support to employees, helping to reduce burnout and promote a healthier work environment. These AI tools are not meant to take over decision-making; rather, they assist managers, health professionals, and HR personnel in noticing early signs of stress. By catching these indicators quickly, they can reach out to staff and offer help, addressing concerns before they develop into bigger problems.

## Figures and Tables

**Figure 1 nursrep-15-00373-f001:**
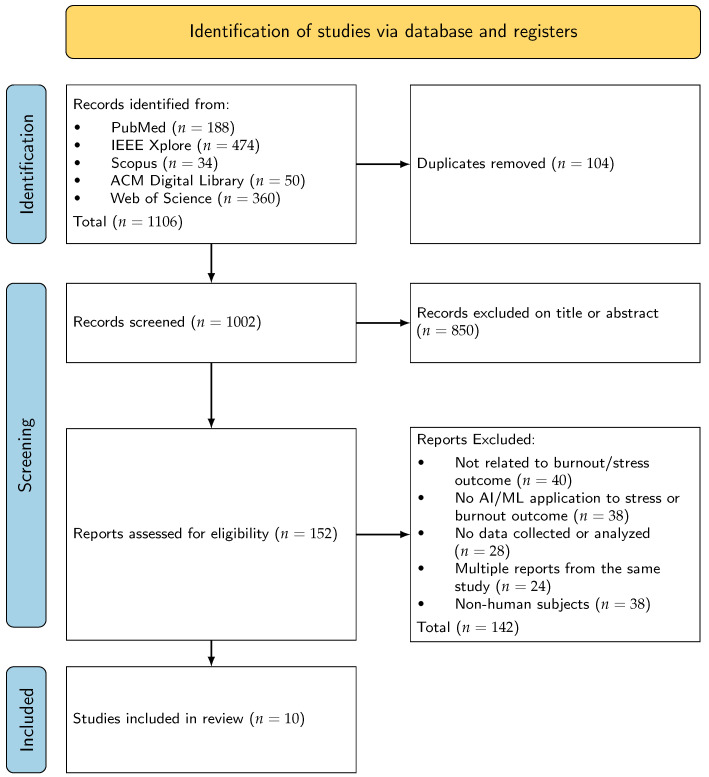
PRISMA Flow Diagram [42].

**Table 1 nursrep-15-00373-t001:** Comparison of Existing Literature Reviews on Stress and Burnout Detection.

Review Paper	Scope: Frontline Workers	Modalities: Workflow Logs, Digital Communications	Key Contributions
Proposed review	✓	✓	Frontline workers in high-stress sectors (healthcare, education, retail, emergency). Comprehensive passive AI synthesis across sectors; identifies key biomarkers (HRV, sleep) and stresses through continuous monitoring. Passive data only: wearables (HR, HRV, EDA, sleep), EHR/workflow logs, digital communication.
Barac et al., 2024 [23]	✓	X	Review of 10 studies; wearable devices measuring heart rate variability (HRV) consistently showed that lower HRV is associated with higher stress and burnout levels in healthcare professionals.
Abd-Alrazaq et al., 2024 [25]	X	X	Meta-analyzed wearable-AI performance in student stress detection, finding a pooled accuracy of 85.6%. Identified factors affecting performance (number of stress classes, device type/location, data size, labeling methods) through subgroup analysis. University students (academic stress). Wearable sensors (HR, HRV, EDA).
Kapogianni et al., 2025 [26]	X	X	Synthesizes 61 studies (2016–2025) on smartwatches for stress/mental health. Finds wearables continuously capture key biomarkers (HRV, EDA) enabling early stress detection and supportive interventions. General public in health/stress contexts. Smartwatch biosensors (HRV, EDA, temp) + behavioral logs.
Pinge et al., 2024 [27]	X	X	In-depth pipeline-focused review; technical method categorization. Individuals under stress (non-sector specific). Wearables and smartphones (HR, EDA, accelerometers).
Ramírez, 2023 [28]	X	X	Scoping review of 40 studies on wearable-based stress management. Reports that most interventions with commercial wearables (smartwatches/bands) yielded significant stress reduction in users. Classifies intervention goals: immediate self-regulation, long-term stress therapy, and stress awareness/education. General population using wearables for stress management. Wearables (HR, GSR); breathing/biofeedback prompts.
Lialiou & Maglogiannis, 2025 [29]	X	X	First systematic review on academic burnout via wearables; shows promise in early burnout symptom prediction. College students (academic burnout). Smartwatches (HR, HRV).
Kargarandehkordi et al., 2024 [30]	X	X	Individuals with stress, anxiety, or depression. Wearables (ECG, PPG, GSR, motion sensors).

Note: Acronyms used in the table include: EHR = Electronic Health Record, SVM = Support Vector Machine, FL = Federated Learning, HR = Heart Rate, HRV = Heart Rate Variability, EDA = Electrodermal Activity, EEG = Electroencephalogram, ECG = Electrocardiogram, EMG = Electromyogram, PPG = Photoplethysmogram, GSR = Galvanic Skin Response, k-NN = k-Nearest Neighbors, LLM = Large Language Model. ✓ indicates that the study is within the scope defined by the column header; X indicates that it is not.

**Table 2 nursrep-15-00373-t002:** Summary of select passive burnout/stress detection studies (2020–2025).

Author, Year & Country	Primary Aim & Study Design	Population & Sample Size (Sector)	Data Source	Model	Key Findings	Quality % (MMAT)
Estévez-Mujica (2018) [43] China	To investigate if e-mail communications can identify risk of burnout Observational study	R&D employees (*n* = 57) Non-healthcare	Email logs	SVM, F1 = 0.84	Communication timing predicted burnout	86% (6/7)
Pinge et al. (2024) [27] India	To review sensors and wearable devices used to detect and monitor stress Systematic review	n/a Non-healthcare	Wearables	Random Forests	Accuracy = 76–95% using sensor data	77% (10/13)
Li et al. (2022) [44] China	To assess workplace stress among nurses using HRV analysis with wearable ECG Pilot experimental study	Nurses *(n* = 30) Healthcare	ECG (HRV)	Statistical analysis	HRV metrics correlated with stress levels	100% (7/7)
Tiase et al. (2024) [45] USA	To conceptualize a logical data model for analyzing nurse–EHR interactions Conceptual framework	Nurses (n/a) Healthcare	EHR logs	Temporal unsupervised classification	Patterns in EHR logs linked to workload and burnout	71% (5/7)
Garcia et al. (2024) [46] USA	To evaluate LLM-generated draft responses to patient messages Randomized controlled trial	Nurses *(n* = 162) Healthcare	EHR inbox logs	GPT-4	AI drafts reduced task load and exhaustion	100% (7/7)
Haghi et al. (2020) [47] Germany	To develop a wrist-worn device for monitoring environmental and physiological parameters Prototype development study	Adults *(n* = 5) Non-healthcare	Wearables	Smartphone app	Device enables stress-related monitoring	100% (7/7)
Giannakakis et al.(2022) [48] Greece	To review biosignal-based methods for psychological stress detection Systematic review	n/a Non-healthcare	EEG, ECG, EDA, EMG, speech, eye movement	SVM, k-NN, RF, Neural Nets	Identified consistent biosignal patterns for stress	54% (7/13)
Fauzi et al. (2022) [49] Norway	To study if FL can improve privacy of smartwatch stress data Experimental study	WESAD dataset *(n* = 15) Non-healthcare	Smartwatch (HR, EDA)	Federated Learning	FL preserved privacy with slight accuracy loss	100% (7/7)
Van Zyl-Cillié (2024) [50] South Africa	To determine if ML models can identify burnout risk Cross -sectional study	Nurses *(n* = 1165) Healthcare	Surveys	Gradient Boosting	Fatigue and support were key predictors (Acc = 75.8%)	100% (7/7)

## Data Availability

No new data were created or analyzed in this study. Data sharing is not applicable to this article.

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
