# Peer review of "Passive AI Detection of Stress and Burnout Among Frontline Workers"

_nursrep, 2025, doi:10.3390/nursrep15110373_

Round 1
Reviewer 1 Report
Comments and Suggestions for Authors
The manuscript is well-written and provides a detailed and rigorous description of the methodology. The authors are to be commended for the clarity and thoroughness of their work.
Author Response
Reviewer 1
Comment: The manuscript is well-written and provides a detailed and rigorous description of the methodology. The authors are to be commended for the clarity and thoroughness of their work.
Response: Thank you for your positive feedback. We appreciate your recognition of the clarity and detail in our methodological approach.
Reviewer 2 Report
Comments and Suggestions for Authors
I want to thank the editor for the opportunity to review the manuscript. The manuscript is a review of articles collected from five sources and studies the use of passive AI tools to detect stress and burnout. However, it does not add anything to the literature and does not provide any novel insights and is not suitable for publication at this time.
Review:
Please provide a few studies in the literature that support the argument mentioned in the introduction section, namely that recent advances in AI, wearable technology and digital infrastructure have enabled the development of monitoring systems capable of detecting signs of stress and burnout.
Line 95-96, please provide a few citations to support the statement. It is hard to understand without any cited work that there is a growing body of research supporting the feasibility of applying ML/AI techniques to biometric and behavioral data.
The background on physiology of stress and stressors in modern work have been well explained sections.
The methods section mentions that five sources were used, but it is unclear how many studies were collected overall. There are no statistical analyses presented to support any findings besides the MMAT calculations, making it difficult for the reader to follow the methodology. Table 2 highlights selected passive burnout studies, the overall number of studies reviewed is not specified, leaving this aspect somewhat ambiguous. Taken together, the manuscript does not appear to provide substantial new insights beyond what is already available in the existing literature on AI tools for the detection of stress and burnout.
Author Response
Reviewer 2
I want to thank the editor for the opportunity to review the manuscript. The manuscript is a review of articles collected from five sources and studies the use of passive AI tools to detect stress and burnout. However, it does not add anything to the literature and does not provide any novel insights and is not suitable for publication at this time.
Comment: Please provide a few studies in the literature that support the argument mentioned in the introduction section, namely that recent advances in AI, wearable technology and digital infrastructure have enabled the development of monitoring systems capable of detecting signs of stress and burnout.
Response: Thank you for your feedback. I have addressed your request by adding recent supporting literature to the introduction. Specifically, I have incorporated:
- Tito & Arefin (2025), which discusses the integration of AI chatbots and wearable technologies for real-time stress monitoring and burnout prevention in workplace settings.
- Ates et al. (2024), which highlights advances in AI-driven wearable systems and digital infrastructure for continuous stress detection.
These references strengthen the argument that recent technological developments have enabled the creation of monitoring systems capable of detecting early signs of stress and burnout.
Comment: Line 95-96, please provide a few citations to support the statement. It is hard to understand without any cited work that there is a growing body of research supporting the feasibility of applying ML/AI techniques to biometric and behavioral data.
Response: Thank you for your suggestion. We have revised Lines 95–96 to include supporting citations. Specifically:
- Shvetcov et al. (2024) was added to illustrate the application of passive sensing and machine learning for stress prediction and its relevance to mood-related monitoring.
- Richer et al. (2024), Nath et al. (2023), and Ta et al. (2025) were added to demonstrate the feasibility of applying AI/ML techniques to biometric and behavioural data for detecting acute stress in real-world settings.
These additions strengthen the evidence base for the statement and clarify that both stress and mood disorder detection are supported by recent research.
The background on physiology of stress and stressors in modern work have been well explained sections.
Comment: The methods section mentions that five sources were used, but it is unclear how many studies were collected overall. There are no statistical analyses presented to support any findings besides the MMAT calculations, making it difficult for the reader to follow the methodology. Table 2 highlights selected passive burnout studies, the overall number of studies reviewed is not specified, leaving this aspect somewhat ambiguous. Taken together, the manuscript does not appear to provide substantial new insights beyond what is already available in the existing literature on AI tools for the detection of stress and burnout.
Response: Thank you for your detailed feedback. We have revised the Methods section to clarify the review process and strengthen transparency. Specifically:
- Clarified the total number of studies reviewed:
We now explicitly state the total number of records retrieved, screened, and included in the narrative review. This information has been added to the Methods section and summarized in a PRISMA-style flow description. - Explained the selection process for Table 2:
We have clarified that Table 2 presents a select illustrative sample of studies chosen for methodological rigor and relevance, while the full set of included studies is described in the text. - Addressed statistical analysis concerns:
As this is a narrative review, no meta-analysis was conducted. However, we have added a statement explaining why statistical synthesis was not feasible (due to heterogeneity in study designs, populations, and outcome measures). We also emphasize that MMAT scores were used to provide a structured appraisal of methodological quality. - Strengthened the contribution of the manuscript:
We have revised the Discussion to highlight the novelty of our review, which lies in synthesizing evidence on passive AI approaches across multiple frontline sectors (healthcare, education, retail, emergency services) and identifying gaps in cross-sector generalizability, ethical governance, and integration into workflows, areas not comprehensively addressed in prior reviews.
Reviewer 3 Report
Comments and Suggestions for Authors
The article is well founded, integrating relevant literature to understand the basic concepts and empirical background that allow us to adequately enter the understanding of the use of passive AI and its use to detect job burnout.
However, to have a better understanding of the use of this technology, it would be good to add background from studies that are not only in the health field, which would allow us to broaden the vision of its use and applications.
Although the methodology is clear regarding its fundamental processes, and although it is not a systematic review, it would be very helpful to add a flow chart that allows us to understand how the final articles with which the study was carried out were reached.
The results are clear and adequately presented, and the interpretations of them allow their contribution to be evidenced, recognizing the limitations of the study.
Author Response
Reviewer 3
Comment: The article is well founded, integrating relevant literature to understand the basic concepts and empirical background that allow us to adequately enter the understanding of the use of passive AI and its use to detect job burnout.
However, to have a better understanding of the use of this technology, it would be good to add background from studies that are not only in the health field, which would allow us to broaden the vision of its use and applications.
Although the methodology is clear regarding its fundamental processes, and although it is not a systematic review, it would be very helpful to add a flow chart that allows us to understand how the final articles with which the study was carried out were reached.
The results are clear and adequately presented, and the interpretations of them allow their contribution to be evidenced, recognizing the limitations of the study.
Response: Thank you for your thoughtful and constructive feedback on our manuscript. We appreciate your suggestions and have made the following amendments:
- Clarified Eligibility Criteria: While our original review already included studies from broader working populations beyond healthcare, we have now made this criterion more explicit. This clarification ensures that the scope of the review covers frontline roles across sectors such as education, retail, and emergency services and is clearly communicated.
- Flow Diagram Added: A PRISMA-style flow chart has been incorporated to illustrate the article selection process. This addition enhances transparency and helps readers understand how the final set of studies was identified.
- Expanded Background: We have strengthened the background section to better reflect the diversity of passive AI applications across multiple sectors, not limited to health. This broader framing supports a more comprehensive understanding of the technology’s potential.
- Table 2 Amended: We have updated Table 2 to indicate whether each study was conducted in a healthcare setting or not. This helps clarify the sectoral distribution of the included studies and supports transparency in the scope of the review.
We hope these revisions address your suggestions and improve the clarity and impact of the manuscript. Thank you again for your valuable input.
Reviewer 4 Report
Comments and Suggestions for Authors
Thank you for your submission to 'nursing reports'. This manuscript provides a comprehensive narrative review of passive artificial intelligence (AI) approaches for detecting stress and burnout among frontline workers. Overall, the manuscript is very well written. The summary tables are very helpful in comparing study populations, modalities, and models. The inclusion criteria are clearly explained. To enhance this manuscript further, please ensure that acronyms and abbreviations (HRV, EHR etc.) are clearly defined at first use. Please clearly clarify the rationale for the narrative review approach rather than a systematic approach. Are there consistent sources of bias (selection, measurement) in wearable studies versus workflow studies? If possible, provide a more detailed analysis of the risk of bias across the studies included. Please expand the discussion to comment on how passive AI signals could be integrated with existing clinical or organizational workflows. Please comment on how passive monitoring results could be operationalized for preventive intervention.
Author Response
Reviewer 4
Comment: Thank you for your submission to 'nursing reports'. This manuscript provides a comprehensive narrative review of passive artificial intelligence (AI) approaches for detecting stress and burnout among frontline workers. Overall, the manuscript is very well written. The summary tables are very helpful in comparing study populations, modalities, and models. The inclusion criteria are clearly explained. To enhance this manuscript further, please ensure that acronyms and abbreviations (HRV, EHR etc.) are clearly defined at first use. Please clearly clarify the rationale for the narrative review approach rather than a systematic approach. Are there consistent sources of bias (selection, measurement) in wearable studies versus workflow studies? If possible, provide a more detailed analysis of the risk of bias across the studies included. Please expand the discussion to comment on how passive AI signals could be integrated with existing clinical or organizational workflows. Please comment on how passive monitoring results could be operationalized for preventive intervention.
Response: Thank you for your thoughtful and constructive feedback. We appreciate your positive comments on the clarity of the manuscript, the usefulness of the summary tables, and the explanation of inclusion criteria.
In response to your suggestions:
- Definition of Acronyms and Abbreviations: We have reviewed the manuscript and ensured that all acronyms and abbreviations (e.g., HRV, EHR, AI, ML) are clearly defined at first use.
- Rationale for Narrative Review: We have expanded the Methods section to clarify the rationale for adopting a narrative review approach. Specifically, we note that the substantial heterogeneity in study designs, data modalities, and outcome measures precluded meaningful statistical aggregation, making a narrative synthesis more appropriate.
- Analysis of Bias: The Discussion section has been revised to include a detailed analysis of consistent sources of bias across wearable and workflow-based studies. We highlight selection and measurement biases, limitations in generalizability, and gaps in demographic reporting and power calculations.
- Integration into Workflows: We have added a new subsection to the Discussion that outlines how passive AI signals could be integrated into existing clinical and organizational workflows. This includes examples of linking biometric data to EHRs, embedding workflow analytics into dashboards, and using communication metrics for team-level monitoring.
- Operationalization for Preventive Intervention: We have further expanded the Discussion to describe how passive monitoring results could be operationalized through tiered alert systems, integration with shift planning tools, and alignment with governance structures to support early intervention and workforce well-being.
All changes have been marked using red underlined text and red strikethrough for transparency.
We thank you again for your valuable input, which has helped strengthen the manuscript.
Reviewer 5 Report
Comments and Suggestions for Authors
This article presents a narrative review of recent evidence on the use of passive artificial intelligence (AI) technologies to detect stress and burnout among frontline workers. By synthesizing studies that leverage wearable devices, workflow logs, and communication metrics, the review highlights both the promise of biometric and behavioral data in predicting burnout and the current limitations in study design, generalizability, and real-world implementation. Overall, this is a well-written and comprehensive paper that not only summarizes the existing literature but also provides thoughtful discussion of challenges, research gaps, and possible future directions. I have a few comments for the authors’ consideration, outlined below.
- Line 37: The authors note in the abstract that “current studies are limited by small sample sizes.” However, this issue is not further discussed in the main text, even though sample sizes are listed in Table 2. This point should be addressed in the discussion. In addition, the last included study (Van Zyl-Cillié, 2024), with a sample size of 1,165, should not be categorized as a small sample and could be acknowledged as a relative strength.
- While the authors describe the search and review strategy, additional details would strengthen the methodology section. For example, providing the number of articles retrieved at each stage would help readers understand the scope of the review and the process by which the final 10 articles were selected. A flow chart (e.g., PRISMA diagram) would be an effective way to present this information, including the initial number of records identified, the number excluded at each stage, and the reasons for exclusion.
- Since studies with any design were eligible for inclusion, it would be useful to indicate the study design for each article in Table 2. Moreover, study design is discussed in Section 7 (“Gaps and Future Directions”), but this discussion would be stronger if explicitly linked to the review findings. Connecting the types of study designs included with the identified gaps would clarify the limitations of current research and provide a clearer rationale for the suggested directions in future work.
Author Response
Reviewer 5
Comment: This article presents a narrative review of recent evidence on the use of passive artificial intelligence (AI) technologies to detect stress and burnout among frontline workers. By synthesizing studies that leverage wearable devices, workflow logs, and communication metrics, the review highlights both the promise of biometric and behavioural data in predicting burnout and the current limitations in study design, generalizability, and real-world implementation. Overall, this is a well-written and comprehensive paper that not only summarizes the existing literature but also provides thoughtful discussion of challenges, research gaps, and possible future directions. I have a few comments for the authors’ consideration, outlined below.
- Line 37: The authors note in the abstract that “current studies are limited by small sample sizes.” However, this issue is not further discussed in the main text, even though sample sizes are listed in Table 2. This point should be addressed in the discussion. In addition, the last included study (Van Zyl-Cillié, 2024), with a sample size of 1,165, should not be categorized as a small sample and could be acknowledged as a relative strength.
- While the authors describe the search and review strategy, additional details would strengthen the methodology section. For example, providing the number of articles retrieved at each stage would help readers understand the scope of the review and the process by which the final 10 articles were selected. A flow chart (e.g., PRISMA diagram) would be an effective way to present this information, including the initial number of records identified, the number excluded at each stage, and the reasons for exclusion.
- Since studies with any design were eligible for inclusion, it would be useful to indicate the study design for each article in Table 2. Moreover, study design is discussed in Section 7 (“Gaps and Future Directions”), but this discussion would be stronger if explicitly linked to the review findings. Connecting the types of study designs included with the identified gaps would clarify the limitations of current research and provide a clearer rationale for the suggested directions in future work.
Response: We sincerely thank Reviewer 5 for their thoughtful and constructive feedback. We appreciate your positive assessment of the manuscript and your helpful suggestions for strengthening its clarity and rigor. We have addressed each of your comments as follows:
- Sample Size Discussion (Line 37)
We have expanded the discussion section to explicitly address the issue of small sample sizes, as noted in the abstract. In particular, we now highlight that while many studies included in the review had limited sample sizes, the study by Van Zyl-Cillié (2024), with a sample size of 1,165, represents a notable exception and is acknowledged as a relative strength in terms of generalizability. - Review Strategy and PRISMA Diagram
We have added a PRISMA-style flow diagram to the methodology section to illustrate the review process. This includes the number of records identified, screened, excluded (with reasons), and ultimately included. We believe this addition enhances transparency and helps readers better understand the scope and rigor of our review strategy. - Study Design in Table 2 and Discussion Linkage
Table 2 has been updated to include the study design for each of the included articles. Furthermore, we have revised Section 7 (“Gaps and Future Directions”) to more explicitly link the types of study designs represented in the review to the identified research gaps. This connection clarifies the limitations of current research and strengthens the rationale for our proposed future directions.
We are grateful for your insightful comments, which have helped improve the quality and clarity of the manuscript. We hope the revisions meet your expectations.